# IMPROVED LANGUAGE MODELING BY DECODING THE PAST

## ABSTRACT

Highly regularized LSTMs achieve impressive results on several benchmark datasets in language modeling. We propose a new regularization method based on decoding the last token in the context using the predicted distribution of the next token. This biases the model towards retaining more contextual information, in turn improving its ability to predict the next token. With negligible overhead in the number of parameters and training time, our Past Decode Regularization (PDR) method achieves a word level perplexity of 55.6 on the Penn Treebank and 63.5 on the WikiText-2 datasets using a single softmax. We also show gains by using PDR in combination with a mixture-of-softmaxes, achieving a word level perplexity of 53.8 and 60.5 on these datasets. In addition, our method achieves 1.169 bits-per-character on the Penn Treebank Character dataset for character level language modeling. These results constitute a new state-of-the-art in their respective settings.

## 1 INTRODUCTION

Language modeling is a fundamental task in natural language processing. Given a sequence of tokens, its joint probability distribution can be modeled using the auto-regressive conditional factorization. This leads to a convenient formulation where a language model has to predict the next token given a sequence of tokens as context. Recurrent neural networks are an effective way to compute distributed representations of the context by sequentially operating on the embeddings of the tokens. These representations can then be used to predict the next token as a probability distribution over a fixed vocabulary using a linear decoder followed by Softmax.

Starting from the work of Mikolov et al. (2010), there has been a long list of works that seek to improve language modeling performance using more sophisticated recurrent neural networks (RNNs) (Zaremba et al. (2014); Zilly et al. (2017); Zoph & Le (2016); Mujika et al. (2017)). However, in more recent work vanilla LSTMs (Hochreiter & Schmidhuber (1997)) with relatively large number of parameters have been shown to achieve state-of-the-art performance on several standard benchmark datasets both in word-level and character-level perplexity (Merity et al. (2018a;b); Melis et al. (2018); Yang et al. (2017)). A key component in these models is the use of several forms of regularization e.g. variational dropout on the token embeddings (Gal & Ghahramani (2016)), dropout on the hidden-to-hidden weights in the LSTM (Wan et al. (2013)), norm regularization on the outputs of the LSTM and classical dropout (Srivastava et al. (2014)). By carefully tuning the hyperparameters associated with these regularizers combined with optimization algorithms like NT-ASGD (a variant of the Averaged SGD), it is possible to achieve very good performance. Each of these regularizations address different parts of the LSTM model and are general techniques that could be applied to any other sequence modeling problem.

In this paper, we propose a regularization technique that is specific to language modeling. One unique aspect of language modeling using LSTMs (or any RNN) is that at each time step $t$, the model takes as input a particular token $x_t$ from a vocabulary $W$ and using the hidden state of the LSTM (which encodes the context till $x_t$) predicts a probability distribution $\mathbf{w}_{t+1}$ on the next token $x_{t+1}$ over the same vocabulary as output. Since $x_t$ can be mapped to a trivial probability distribution over $W$, this operation can be interpreted as transforming distributions over $W$ (Inan et al. (2016)). Clearly, the output distribution is dependent on and is a function of $x_t$ and the context further in the past and encodes information about it. We ask the following question – How much information

is it possible to decode about the input distribution (and hence $x_t$) from the output distribution $\mathbf{w}_{t+1}$? In general, it is impossible to decode $x_t$ unambiguously. Even if the language model is perfect and correctly predicts $x_{t+1}$ with probability 1, there could be many tokens preceding it. However, in this case the number of possibilities for $x_t$ will be limited, as dictated by the bigram statistics of the corpus and the language in general. We argue that biasing the language model such that it is possible to decode more information about the past tokens from the predicted next token distribution is beneficial. We incorporate this intuition into a regularization term in the loss function of the language model.

The symmetry in the inputs and outputs of the language model at each step lends itself to a simple decoding operation. It can be cast as a (pseudo) language modeling problem in "reverse", where the future prediction $\mathbf{w}_{t+1}$ acts as the input and the last token $x_t$ acts as the target of prediction. The token embedding matrix and weights of the linear decoder of the main language model can be reused in the past decoding operation. We only need a few extra parameters to model the nonlinear transformation performed by the LSTM, which we do by using a simple stateless layer. We compute the cross-entropy loss between the decoded distribution for the past token and $x_t$ and add it to the main loss function after suitable weighting. The extra parameters used in the past decoding are discarded during inference time. We call our method *Past Decode Regularization* or **PDR** for short.

We conduct extensive experiments on four benchmark datasets for word level and character level language modeling by combining PDR with existing LSTM based language models and achieve new state-of-the-art performance on three of them.

## 2 PAST DECODE REGULARIZATION (PDR)

Let $\mathbf{X} = (x_1, x_2, \cdots, x_t, \cdots, x_T)$ be a sequence of tokens. In this paper, we will experiment with both word level and character level language modeling. Therefore, tokens can be either words or characters. The joint probability $P(\mathbf{X})$ factorizes into

$$P(\mathbf{X}) = \prod_{t=1}^{T} P(x_t | x_1, x_2, \cdots, x_{t-1}) \tag{1}$$

Let $c_t = (x_1, x_2, \cdots, x_t)$ denote the context available to the language model for $x_{t+1}$. Let $W$ denote the vocabulary of tokens, each of which is embedded into a vector of dimension $d$. Let $\mathbf{E}$ denote the token embedding matrix of dimension $|W| \times d$ and $\mathbf{e}_w$ denote the embedding of $w \in W$. An LSTM computes a distributed representation of $c_t$ in the form of its hidden state $\mathbf{h}_t$, which we assume has dimension $d$ as well. The probability that the next token is $w$ can then be calculated using a linear decoder followed by a Softmax layer as

$$P_\theta(w|c_t) = \text{Softmax}(\mathbf{h}_t \mathbf{E}^{\text{T}} + \mathbf{b})|_w = \frac{\exp(\mathbf{h}_t \mathbf{e}_w^{\text{T}})}{\sum_{w' \in W} \exp(\mathbf{h}_t \mathbf{e}_{w'}^{\text{T}} + b_{w'})} \tag{2}$$

where $b_{w'}$ is the entry corresponding to $w'$ in a bias vector $\mathbf{b}$ of dimension $|W|$ and $|_w$ represents projection onto $w$. Here we assume that the weights of the decoder are tied with the token embedding matrix $\mathbf{E}$ (Inan et al. (2016); Press & Wolf (2017)). To optimize the parameters of the language model $\theta$, the loss function to be minimized during training is set as the cross-entropy between the predicted distribution $P_\theta(w|c_t)$ and the actual token $x_{t+1}$.

$$\mathcal{L}_{CE} = \sum_t -\log(P_\theta(x_{t+1}|c_t)) \tag{3}$$

Note that Eq.(2), when applied to all $w \in W$ produces a $1 \times |W|$ vector $\mathbf{w}_{t+1}$, encapsulating the prediction the language model has about the next token $x_{t+1}$. Since this is dependent on and conditioned on $c_t$, $\mathbf{w}_{t+1}$ clearly encodes information about it; in particular about the last token $x_t$ in $c_t$. In turn, it should be possible to infer or decode some limited information about $x_t$ from $\mathbf{w}_{t+1}$. We argue that by biasing the model to be more accurate in recalling information about past tokens, we can help it in predicting the next token better.

To this end, we define the following decoding operation to compute a probability distribution over $w_c \in W$ as the last token in the context.

$$P_{\theta_r}(w_c | \mathbf{w}_{t+1}) = \text{Softmax}(f_{\theta_r}(\mathbf{w}_{t+1} \mathbf{E}) \mathbf{E}^{\text{T}} + \mathbf{b}'_{\theta_r}) \tag{4}$$

| | **PTB** | | | **WT2** | | | **PTBC** | | | **enwik8** | | |
|---|---|---|---|---|---|---|---|---|---|---|---|---|
| | Train | Valid | Test | Train | Valid | Test | Train | Valid | Test | Train | Valid | Test |
| Tokens | 888K | 70.4K | 78.7K | 2.05M | 213K | 241K | 5.01M | 393k | 442k | 90M | 5M | 5M |
| Vocab | | 10K | | | 33.3K | | | 51 | | | 205 | |

Table 1: Statistics of the language modeling benchmark datasets.

Here $f_{\theta_r}$ is a non-linear function that maps vectors in $\mathbb{R}^d$ to vectors in $\mathbb{R}^d$ and $\mathbf{b}'_{\theta_r}$ is a bias vector of dimension $|W|$, together with parameters $\theta_r$. In effect, we are *decoding the past* – the last token in the context $x_t$. This produces a vector $\mathbf{w}_t^r$ of dimension $1 \times |W|$. The cross-entropy loss with respect to the actual last token $x_t$ can then be computed as

$$\mathcal{L}_{PDR} = \sum_t -\log(P_{\theta_r}(x_t|\mathbf{w}_{t+1})) \tag{5}$$

Here $PDR$ stands for *Past Decode Regularization*. $\mathcal{L}_{PDR}$ captures the extent to which the decoded distribution of tokens differs from the actual tokens $x_t$ in the context. Note the symmetry between Eqs.(2) and (5). The "input" in the latter case is $\mathbf{w}_{t+1}$ and the "context" is provided by a nonlinear transformation of $\mathbf{w}_{t+1}\mathbf{E}$. Different from the former, the context in Eq.(5) does not preserve any state information across time steps as we want to decode only using $\mathbf{w}_{t+1}$. The term $\mathbf{w}_{t+1}\mathbf{E}$ can be interpreted as a "soft" token embedding lookup, where the token vector $\mathbf{w}_{t+1}$ is a probability distribution instead of a unit vector.

We add $\lambda_{PDR}\mathcal{L}_{PDR}$ to the loss function in Eq.(3) as a regularization term, where $\lambda_{PDR}$ is a positive weighting coefficient, to construct the following new loss function for the language model.

$$\mathcal{L} = \mathcal{L}_{CE} + \lambda_{PDR}\mathcal{L}_{PDR} \tag{6}$$

Thus equivalently PDR can also be viewed as a method of defining an augmented loss function for language modeling. The choice of $\lambda_{PDR}$ dictates the degree to which we want the language model to incorporate our inductive bias i.e. decodability of the last token in the context. If it is too large, the model will fail to predict the next token, which is its primary task. If it is zero or too small, the model will retain less information about the last token which hampers its predictive performance. In practice, we choose $\lambda_{PDR}$ by a search based on validation set performance.

Note that the trainable parameters $\theta_r$ associated with PDR are used only during training to bias the language model and are not used at inference time. This also means that it is important to control the complexity of the nonlinear function $f_{\theta_r}$ so as not to overly bias the training. As a simple choice, we use a single fully connected layer of size $d$ followed by a Tanh nonlinearity as $f_{\theta_r}$. This introduces few extra parameters and a small increase in training time as compared to a model not using PDR.

## 3 EXPERIMENTS

We present extensive experimental results to show the efficacy of using PDR for language modeling on four standard benchmark datasets – two each for word level and character level language modeling. For the former, we evaluate our method on the Penn Treebank (PTB) (Mikolov et al. (2010)) and the WikiText-2 (WT2) (Merity et al. (2016)) datasets. For the latter, we use the Penn Treebank Character (PTBC) (Mikolov et al. (2010)) and the Hutter Prize Wikipedia Prize (Hutter (2018)) (also known as Enwik8) datasets. Key statistics for these datasets is presented in Table 1.

As mentioned in the introduction, some of the best existing results on these datasets are obtained by using extensive regularization techniques on relatively large LSTMs (Merity et al. (2018a;b); Yang et al. (2017)). We apply our regularization technique to these models, the so called AWD-LSTM. We consider two versions of the model – one with a single softmax (AWD-LSTM) and one with a mixture-of-softmaxes (AWD-LSTM-MoS). The PDR regularization term is computed according to Eq.(4) and Eq.(5). We call our model AWD-LSTM+PDR when using a single softmax and AWD-LSTM-MoS+PDR when using a mixture-of-softmaxes. We largely follow the experimental procedure of the original models and incorporate their dropouts and regularizations in our experiments. The relative contribution of these existing regularizations and PDR will be analyzed in Section 6.

There are 7 hyperparameters associated with the regularizations used in AWD-LSTM (and one extra with MoS). PDR also has an associated weighting coefficient $\lambda_{PDR}$. For our experiments, we set $\lambda_{PDR} = 0.001$ which was determined by a coarse search on the PTB and WT2 validation sets. For the remaining ones, we perform light hyperparameter search in the vicinity of those reported for AWD-LSTM in Merity et al. (2018a;b) and for AWD-LSTM-MoS in Yang et al. (2017).

### 3.1 Model and training for PTB and WikiText-2

For the single softmax model (AWD-LSTM+PDR), for both PTB and WT2, we use a 3-layered LSTM with 1150, 1150 and 400 hidden dimensions. The word embedding dimension is set to $d = 400$. For the mixture-of-softmax model, we use a 3-layer LSTM with dimensions 960, 960 and 620, embedding dimension of 280 and 15 experts for PTB and a 3-layer LSTM with dimensions 1150, 1150 and 650, embedding dimension of $d = 300$ and 15 experts for WT2. Weight tying is used in all the models. For training the models, we follow the same procedure as AWD-LSTM i.e. a combination of SGD and NT-ASGD, followed by finetuning. We adopt the learning rate schedules and batch sizes of Merity et al. (2018a) and Yang et al. (2017) in our experiments.

### 3.2 Model and training for PTBC and Enwik8

For PTBC, we use a 3-layer LSTM with 1000, 1000 and 200 hidden dimensions and a character embedding dimension of $d = 200$. For Enwik8, we use a LSTM with 1850, 1850 and 400 hidden dimensions and the characters are embedded in $d = 400$ dimensions. For training, we largely follow the procedure laid out in Merity et al. (2018b). For each of the datasets, AWD-LSTM+PDR has less than 1% more parameters than the corresponding AWD-LSTM model (during training only). The maximum observed time overhead due to the additional computation is less than 3%.

## 4 Results on Word Level Language Modeling

The results for PTB are shown in Table 2. With a single softmax, our method (AWD-LSTM+PDR) achieves a perplexity of 55.6 on the PTB test set, which improves on the current state-of-the-art with a single softmax by an absolute 1.7 points. The advantages of better information retention due to PDR are maintained when combined with a continuous cache pointer (Grave et al. (2016)), where our method yields an absolute improvement of 1.2 over AWD-LSTM. Notably, when coupled with dynamic evaluation (Krause et al. (2018)), the perplexity is decreased further to 49.3. To the best of our knowledge, ours is the first method to achieve a sub 50 perplexity on the PTB test set with a single softmax. Note that, for both cache pointer and dynamic evaluation, we coarsely tune the associated hyperparameters on the validation set.

Using a mixture-of-softmaxes, our method (AWD-LSTM-MoS+PDR) achieves a test perplexity of 53.8, an improvement of 0.6 points over the current state-of-the-art. The use of dynamic evaluation pushes the perplexity further down to 47.3. PTB is a restrictive dataset with a vocabulary of 10K words. Achieving good perplexity requires considerable regularization. The fact that PDR can improve upon existing heavily regularized models is empirical evidence of its distinctive nature and its effectiveness in improving language models.

Table 3 shows the perplexities achieved by our model on WT2. This dataset is considerably more complex than PTB with a vocabulary of more than 33K words. AWD-LSTM+PDR improves over the current state-of-the-art with a single softmax by a significant 2.3 points, achieving a perplexity of 63.5. The gains are maintained with the use of cache pointer (2.4 points) and with the use of dynamic evaluation (1.7 points). Using a mixture-of-softmaxes, AWD-LSTM-MoS+PDR achieves perplexities of 60.5 and 40.3 (with dynamic evaluation) on the WT2 test set, improving upon the current state-of-the-art by 1.0 and 0.4 points respectively.

### 4.1 Performance on Larger Datasets

We consider the Gigaword dataset Chelba et al. (2014) with a truncated vocabulary of about 100K tokens with the highest frequency and apply PDR to a baseline 2-layer LSTM language model with embedding and hidden dimensions set to 1024. We use all the shards from the training set for training and a few shards from the heldout set for validation (heldout-0,10) and test (heldout-20,30,40). We

| Model | #Params | Valid | Test |
|---|---|---|---|
| Sate-of-the-art Methods (Single Softmax) | | | |
| Merity et al. (2018a) – AWD-LSTM | 24.2M | 60.0 | 57.3 |
| Merity et al. (2018a) – AWD-LSTM + continuous cache pointer | 24.2M | 53.9 | 52.8 |
| Krause et al. (2018) – AWD-LSTM + dynamic evaluation | 24.2M | 51.6 | 51.1 |
| Our Method (Single Softmax) | | | |
| AWD-LSTM+PDR | 24.2M | **57.9** | **55.6** (**-1.7**) |
| AWD-LSTM+PDR + continuous cache pointer | 24.2M | 52.4 | 51.6 (**-1.2**) |
| AWD-LSTM+PDR + dynamic evaluation | 24.2M | **50.1** | **49.3** (**-1.8**) |
| Sate-of-the-art Methods (Mixture-of-Softmax) | | | |
| Yang et al. (2017) – AWD-LSTM-MoS | 22M | 56.5 | 54.4 |
| Yang et al. (2017) – AWD-LSTM-MoS + dynamic evaluation | 22M | 48.3 | 47.7 |
| Our Method (Mixture-of-Softmax) | | | |
| AWD-LSTM-MoS+PDR | 22M | **56.2** | **53.8** (**-0.6**) |
| AWD-LSTM-MoS+PDR + dynamic evaluation | 22M | **48.0** | **47.3** (**-0.4**) |

Table 2: Perplexities on Penn Treebank (PTB) test set for single softmax and mixture-of-softmaxes models. Values in parentheses show improvement over respective state-of-the-art perplexities.

| Model | #Params | Valid | Test |
|---|---|---|---|
| Sate-of-the-art Methods (Single Softmax) | | | |
| Merity et al. (2018a) – AWD-LSTM | 33.6M | 68.6 | 65.8 |
| Merity et al. (2018a) – AWD-LSTM + continuous cache pointer | 33.6M | 53.8 | 52.0 |
| Krause et al. (2018) – AWD-LSTM + dynamic evaluation | 33.6M | 46.4 | 44.3 |
| Our Method (Single Softmax) | | | |
| AWD-LSTM+PDR | 33.6M | **66.5** | **63.5** (**-2.3**) |
| AWD-LSTM+PDR + continuous cache pointer | 33.6M | 51.5 | 49.6 (**-2.4**) |
| AWD-LSTM+PDR + dynamic evaluation | 33.6M | **44.6** | **42.6** (**-1.7**) |
| Sate-of-the-art Methods (Mixture-of-Softmax) | | | |
| Yang et al. (2017) – AWD-LSTM-MoS | 35M | 63.9 | 61.5 |
| Yang et al. (2017) – AWD-LSTM-MoS + dynamic evaluation | 35M | 42.4 | 40.7 |
| Our Method (Mixture-of-Softmax) | | | |
| AWD-LSTM-MoS+PDR | 35M | **63.0** | **60.5** (**-1.0**) |
| AWD-LSTM-MoS+PDR + dynamic evaluation | 35M | **42.0** | **40.3** (**-0.4**) |

Table 3: Perplexities on WikiText-2 (WT2) test set for single softmax and mixture-of-softmaxes models. Values in parentheses show improvement over respective state-of-the-art perplexities.

tuned the PDR coefficient coarsely in the vicinity of 0.001. While the baseline model achieved a validation (test) perplexity of 44.3 (43.1), on applying PDR, the model achieved a perplexity of 44.0 (42.5). Thus, PDR is relatively less effective on larger datasets, a fact also observed for other regularization techniques on such datasets (Yang et al. (2017)).

## 5 RESULTS ON CHARACTER LEVEL LANGUAGE MODELING

The results on PTBC are shown in Table 4. Our method achieves a bits-per-character (BPC) performance of 1.169 on the PTBC test set, improving on the current state-of-the-art by 0.006 or 0.5%. It is notable that even with this highly processed dataset and a small vocabulary of only 51 tokens, our method improves on already highly regularized models. Finally, we present results on Enwik8 in

| Model | #Params | Test |
|---|---|---|
| Krueger et al. (2016) – Zoneout LSTM | - | 1.27 |
| Chung et al. (2016) – HM-LSTM | - | 1.24 |
| Ha et al. (2016) – HyperLSTM | 14.4M | 1.219 |
| Zoph & Le (2016) – NAS Cell | 16.3M | 1.214 |
| Mujika et al. (2017) – FS-LSTM-4 | 6.5M | 1.193 |
| Merity et al. (2018b) – AWD-LSTM | 13.8M | 1.175 |
| Our Method | | |
| AWD-LSTM+PDR | 13.8M | 1.169 (**-0.006**) |

Table 4: Bits-per-character on the PTBC test set.

| Model | #Params | Test |
|---|---|---|
| Ha et al. (2016) – HyperLSTM | 27M | 1.340 |
| Chung et al. (2016) – HM-LSTM | 35M | 1.32 |
| Rocki et al. (2016) – SD Zoneout | 64M | 1.31 |
| Zilly et al. (2017) – RHN (depth 10) | 21M | 1.30 |
| Zilly et al. (2017) – Large RHN | 46M | 1.270 |
| Mujika et al. (2017) – FS-LSTM-4 | 27M | 1.277 |
| Mujika et al. (2017) – Large FS-LSTM-4 | 47M | 1.245 |
| Merity et al. (2018b) – AWD-LSTM | 47M | 1.232 |
| Our Method | | |
| AWD-LSTM (Ours) | 47M | 1.257 |
| AWD-LSTM+PDR | 47M | 1.245 (-0.012) |

Table 5: Bits-per-character on Enwik8 test set.

Table 5. AWD-LSTM+PDR achieves 1.245 BPC. This is 0.012 or about 1% less than the 1.257 BPC achieved by AWD-LSTM in our experiments (with hyperparameters from Merity et al. (2018b)).

## 6 ANALYSIS OF PDR

In this section, we analyze PDR by probing its performance in several ways and comparing it with current state-of-the-art models that do not use PDR.

### 6.1 A VALID REGULARIZATION

| | PTB Valid | WT2 Valid |
|---|---|---|
| AWD-LSTM (NoReg) | 108.6 | 142.7 |
| AWD-LSTM (NoReg) + PDR | 106.2 | 137.6 |

Table 6: Validation perplexities for AWD-LSTM without any regularization and with only PDR.

To verify that indeed PDR can act as a form of regularization, we perform the following experiment. We take the models for PTB and WT2 and turn off all dropouts and regularization and compare its performance with only PDR turned on. The results, as shown in Table 6, validate the premise of PDR. The model with only PDR turned on achieves 2.4 and 5.1 better validation perplexity on PTB and WT2 as compared to the model without any regularization. Thus, biasing the LSTM by decoding the distribution of past tokens from the predicted next-token distribution can indeed act as a regularizer leading to better generalization performance.

Next, we plot histograms of the negative log-likelihoods of the correct context tokens $x_t$ in the past decoded vector $\mathbf{w}_t^r$ computed using our best models on the PTB and WT2 validation sets in Fig.

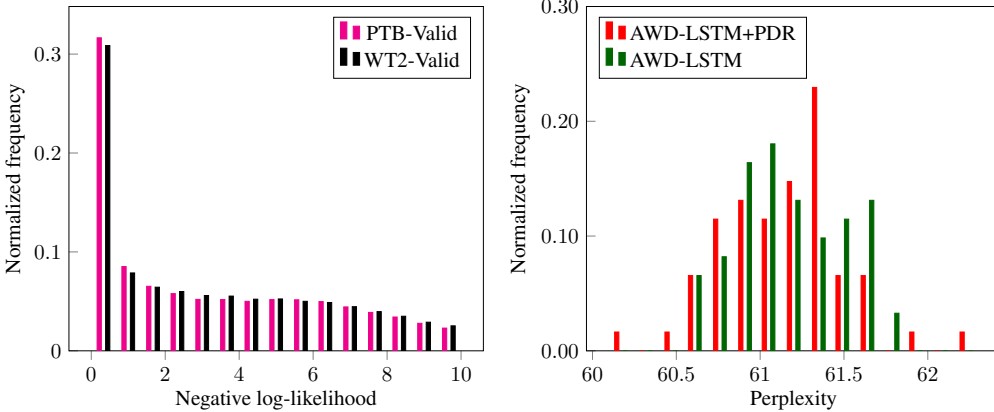

(a) Histogram of the NLL of $x_t$ in the past decoded vector $\mathbf{w}_t^\mathcal{T}$.

(b) Histogram of validation perplexities on PTB for a set of different hyperparameters.

Figure 1: Context token NLL for AWD-LSTM+PDR and comparison with AWD-LSTM.

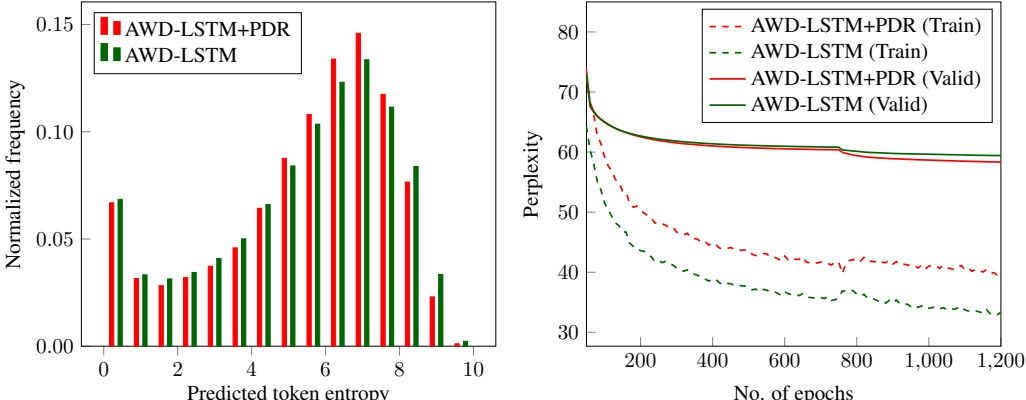

(a) Histogram of entropies of $\mathbf{w}_{t+1}$ for PTB valid.

(b) Training curves on PTB showing perplexity. The kink in the middle represents the start of finetuning.

Figure 2: Comparison between AWD-LSTM+PDR and AWD-LSTM.

1(a). The NLL values are significantly peaked near 0, which means that the past decoding operation is able to decode significant amount of information about the last token in the context.

To investigate the effect of hyperparameters on PDR, we pick 60 sets of random hyperparameters in the vicinity of those reported by Merity et al. (2018a) and compute the validation set perplexity after training (without finetuning) on PTB, for both AWD-LSTM+PDR and AWD-LSTM. Their histograms are plotted in Fig.1(b). The perplexities for models with PDR are distributed slightly to the left of those without PDR. There appears to be more instances of perplexities in the higher range for models without PDR. Note that there are certainly hyperparameter settings where adding PDR leads to lower validation complexity, as is generally the case for any regularization method.

## 6.2 COMPARISON WITH AWD-LSTM

To show the qualitative difference between AWD-LSTM+PDR and AWD-LSTM, in Fig.2(a), we plot a histogram of the entropy of the predicted next token distribution $\mathbf{w}_{t+1}$ for all the tokens in the validation set of PTB achieved by their respective best models. The distributions for the two models is slightly different, with some identifiable patterns. The use of PDR has the effect of reducing the entropy of the predicted distribution when it is in the higher range of 8 and above, pushing it into the range of 5-8. This shows that one way PDR biases the language model is by reducing the entropy of the predicted next token distribution. Indeed, one way to reduce the cross-entropy between $x_t$ and

| Model | PTB | | WT2 | |
|---|---|---|---|---|
| | Valid | Test | Valid | Test |
| AWD-LSTM+PDR | 57.9 | 55.6 | 66.5 | 63.5 |
| – finetune | 60.4 | 58.0 | 68.5 | 65.6 |
| – LSTM output dropout | 67.6 | 65.4 | 75.4 | 72.1 |
| – LSTM layer dropout | 68.1 | 65.8 | 73.7 | 70.4 |
| – embedding dropout | 63.9 | 61.4 | 77.1 | 73.6 |
| – word dropout | 62.9 | 60.5 | 70.4 | 67.4 |
| – LSTM weight dropout | 68.4 | 65.8 | 79.0 | 75.5 |
| – alpha/beta regularization | 63.0 | 60.4 | 74.0 | 70.7 |
| – weight decay | 64.7 | 61.4 | 72.5 | 68.9 |
| – past decoding regularization (PDR) | 60.5 | 57.7 | 69.5 | 66.4 |

Table 7: Ablation experiments on the PTB and WT2 validation sets.

$\mathbf{w}_t^r$ is by making $\mathbf{w}_{t+1}$ less spread out in Eq.(5). This tends to benefits the language model when the predictions are correct.

We also compare the training curves for the two models in Fig.2(b) on PTB. Although the two models use slightly different hyperparameters, the regularization effect of PDR is apparent with a lower validation perplexity but higher training perplexity. The corresponding trends shown in Fig.2(a,b) for WT2 have similar characteristics.

### 6.3 ABLATION STUDIES

We perform a set of ablation experiments on the best AWD-LSTM+PDR models for PTB and WT2 to understand the relative contribution of PDR and the other regularizations used in the model. The results are shown in Table 7. In both cases, PDR has a significant effect in decreasing the validation set performance, albeit lesser than the other forms of regularization. This is not surprising as PDR does not influence the LSTM directly.

## 7 RELATED WORK

Our method builds on the work of using sophisticated regularization techniques to train LSTMs for language modeling. In particular, the AWD-LSTM model achieves state-of-the-art performance with a single softmax on the four datasets considered in this paper (Merity et al. (2018a;b)). Melis et al. (2018) also achieve similar results with highly regularized LSTMs. By addressing the so-called softmax bottleneck in single softmax models, Yang et al. (2017) use a mixture-of-softmaxes to achieve significantly lower perplexities. PDR utilizes the symmetry between the inputs and outputs of a language model, a fact that is also exploited in weight tying (Inan et al. (2016); Press & Wolf (2017)). Our method can be used with untied weights as well. Although motivated by language modeling, PDR can also be applied to seq2seq models with shared input-output vocabularies, such as those used for text summarization and neural machine translation (with byte pair encoding of words) (Press & Wolf (2017)). Regularizing the training of an LSTM by combining the main objective function with auxiliary tasks has been successfully applied to several tasks in NLP (Radford et al. (2018); Rei (2017)). In fact, a popular choice for the auxiliary task is language modeling itself. This in turn is related to multi-task learning (Collobert & Weston (2008)).

Specialized architectures like Recurrent Highway Networks (Zilly et al. (2017)) and NAS (Zoph & Le (2016)) have been successfully used to achieve competitive performance in language modeling. The former one makes the hidden-to-hidden transition function more complex allowing for more refined information flow. Such architectures are especially important for character level language modeling where strong results have been shown using Fast-Slow RNNs (Mujika et al. (2017)), a two level architecture where the slowly changing recurrent network tries to capture more long range dependencies. The use of historical information can greatly help language models deal with long range dependencies as shown by Merity et al. (2016); Krause et al. (2018); Rae et al. (2018). Finally, in a recent paper, Gong et al. (2018) achieve improved performance for language modeling by using frequency agnostic word embeddings, a technique orthogonal to and combinable with PDR.

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
