# OpenReview forum: "Improved Language Modeling by Decoding the Past"
_ICLR.cc/2019/Conference_

### Official Review · AnonReviewer2 · 2018-10-25
**At best misleading notion of state-of-the-art, optimistic evaluation**

**Rating:** 3
**Confidence:** 5

**Review:**

In their abstract, the authors claim to provide state-of-the-art perplexity on Penn Treebank, which is not true. As the authors state, their notion of "state-of-the-art" excludes exactly that earlier work, which does provide state-of-the-art perplexity on Penn Treebank (Yang et al. 2017), as stated in Sec. 4.1. The question is, why one would exlude the mixture-of-softmax approach here? This is clearly misleading.

The authors introduce the idea of past decoding for the purpose of regularization. It remains somewhat unclear, why this bigram-centered regularization would strongly contribute for prediction in general.

The results obtained show moderate improvements of approx. 1 point in perplexity on top of their best current result on Penn Treebank. Considering the small size of the corpus for the evaluation of a regularization method, the results even seem optimistic - it remains unclear, if this approach would readily scale to larger datasets. The mode of language modeling evaluation presented here, without considering an actual language or speech processing task, provides limited insight w.r.t. its utility in actual applications. Moreover, the very limited size of the language modeling tasks chosen here is highly advantageous for smoothing/regularization approaches. It remains totally unclear, how the presented approaches would perform on more realistically sized tasks and within actual applications.

---

> ### Author Response · Authors · 2018-11-20
> **Re: At best misleading notion of state-of-the-art, optimistic evaluation**
>
> We thank the reviewer for reading the paper and the comments. As already stated in the comments below, our claim of state-of-the-art in the original manuscript pertains to models with a single softmax, which we clearly state in section 4.1. We will update the abstract to remove any confusion. As suggested by multiple reviewers, we have performed further experiments by incorporating our Past Decode Regularization (PDR) in the mixture-of-softmax (AWD-LSTM-MoS) model of (Yang et al. 2017). We use the same model sizes as used in the paper. As shown below, we observe gains of 0.4 and 1.0 perplexity points for PTB and WT2, while with dynamic evaluation the gains are 0.4 in both cases.
>
>                                                         AWD-LSTM-MoS+PDR  || AWD-LSTM-MoS (Yang et al. 2017)
> Penn Treebank with finetuning -                   56.2/53.8   ||  56.5/54.4
> Penn Treebank with dynamic evaluation -   48.0/47.3   ||  48.3/47.7
>
> WikiText-2 with finetuning -                            63.0/60.5   ||  63.9/61.5
> WikiText-2 with dynamic evaluation -            42.0/40.3   ||  42.4/40.7
>
> Note that, we performed very limited hyperparameter tuning in the vicinity of the hyperparameters used by (Yang et al. 2017) and a more exhaustive search is likely to lead to better gains. Thus, the gains due to PDR generalize to more complex models like AWD-LSTM-MoS+PDR.
>
> We can justify PDR theoretically as an inductive bias on the language model. The observed bigrams in a language are not random and the distribution of the second word given the first word in a bigram is not uniform. Similarly, the distribution of the first word given the second word will be far from uniform. A RNN based language model models the first dependence (and more long range ones) and our proposed PDR tries to model the second one. In a unidirectional language model, we cannot look into the future tokens and hence we use the output distribution as a proxy for the "true second word" and decode the distribution of the first word. Thus the PDR term can be thought of as biasing the language model to retain more information about the distribution of the first word given the second word in a bigram.
>
> We believe language modeling is a fundamental problem in NLP and our work continues a long stream of papers that have achieved steadily lower perplexities over the past few years. We evaluated our approach on two standard datasets that have been used as a benchmark in most of these papers.
>
> As suggested by multiple reviewers, we have conducted further experiments on the Gigaword corpus to test PDR on larger corpora. Specifically,  we use a 2-layer LSTM with hidden dimension 1024 and a word embedding dimension of 1024. We truncated the vocabulary by keeping approximately 100k words with the highest frequency and used the same validation and test sets as (Yang et al. 2017). We obtained a valid/test perplexity of 44.0/42.5 for the model with PDR and 44.3/43.1 for the model without PDR, showing a gain of 0.6 points in the test perplexity. Note that we tuned the PDR loss coefficient very coarsely and tuning it further could lead to higher gains. We will update the manuscript with these additional results and discussion and post it shortly.
>
> Yang et al. 2017. Breaking the softmax bottleneck: A high-rank RNN language model. arXiv:1711.03953.

---

> > ### Comment · AnonReviewer2 · 2018-11-21
> > **Re: Re: At best misleading notion of state-of-the-art, optimistic evaluation**
> >
> > I did understand your point about excluding MoS in your results. However, I do not agree to the notion of "state-of-the-art" under constraints that explicitly exclude actual state-of-the-art from the discussion.
> >
> > However, it is good to see additional experiments being done on top of current state-of-the-art results. The authors now show in addition that also on top of MoS results, consistent, though very limited improvements of less then one point in perplexity are obtained. Also, results on the much larger Gigaword corpus are presented, where PDR shows, not unexpected, an even smaller gain in perplexity of only 0.3 - notwithstanding the proposed further hyperparameter optimization.
> >
> > Working intensely on applications where language models contribute strongly, I sincerely doubt that minor improvements in perplexity like those reported here will transfer to respective applications give significant performance improvements - I have seen much larger perplexity gains fail to transfer repeatedly.

---

### Official Review · AnonReviewer3 · 2018-10-28
**A useful regularization for RNN language models**

**Rating:** 7
**Confidence:** 5

**Review:**

The paper suggests a new regularization technique which can be added on top of those used in AWD-LSTM of Merity et al. (2017) with little overhead.

This is a well-written paper with a clear structure. The experiments are presented in a clear and understandable fashion, and the evaluation seems thorough. The methodology seems sound, and the authors present the reader with all the information needed to replicate the experiments.

I would only suggest evaluating this technique on AWD-LSTM-MoS of Yang et al. (2017) to get a more complete picture.

References
- Merity, S., Keskar, N.S. and Socher, R., 2017. Regularizing and optimizing LSTM language models. arXiv preprint arXiv:1708.02182.
- Yang, Z., Dai, Z., Salakhutdinov, R. and Cohen, W.W., 2017. Breaking the softmax bottleneck: A high-rank RNN language model. arXiv preprint arXiv:1711.03953.

---

> ### Author Response · Authors · 2018-11-20
> **Re: A useful regularization for RNN language models**
>
> We thank the reviewer very much for reading the paper carefully and providing us with constructive comments. We have conducted further experiments applying our Past Decode Regularization (PDR) to the mixture-of-softmax (AWD-LSTM-MoS) model of (Yang et al. 2017). We use the same model sizes as in the paper. Even with the very limited hyperparameter search in the vicinity of those used in the paper and fixing the PDR loss coefficient to 0.001 (as used in the other models in our paper), we see consistent gains on the Penn Treebank and WikiText-2 datasets. The validation/test perplexities are as follows -
>
>                                                         AWD-LSTM-MoS+PDR  || AWD-LSTM-MoS (Yang et al. 2017)
> Penn Treebank with finetuning -                   56.2/53.8   ||  56.5/54.4
> Penn Treebank with dynamic evaluation -   48.0/47.3   ||  48.3/47.7
>
> WikiText-2 with finetuning -                            63.0/60.5   ||  63.9/61.5
> WikiText-2 with dynamic evaluation -            42.0/40.3   ||  42.4/40.7
>
> Thus we observe gains of 0.6 and 1.0 points in test perplexity for PTB and WT2. With dynamic evaluation, the gains for both datasets is 0.4 points. Note again that we did a very limited hyperparameter search and more exhaustive experiments will likely lead to even better gains by using PDR. We will update and reorganize the experiments section in the paper accordingly. The updated manuscript will be posted shortly.
>
> Yang et al. 2017. Breaking the softmax bottleneck: A high-rank RNN language model. arXiv:1711.03953.

---

> > ### Comment · AnonReviewer3 · 2018-12-10
> > **Thanks for the update**
> >
> > I am sorry for the late reply, and thank you for the update. I am convinced that the suggested technique is useful.

---

### Official Review · AnonReviewer1 · 2018-11-02
**Weak accept**

**Rating:** 6
**Confidence:** 3

**Review:**

This paper proposes an additional loss term to use when training an LSTM LM.  The authors argue that, intuitively, we want the output distribution to retain some information about the context, or "past".  Given this, they use the output distribution as input to a one layer network that must predict the current token.  The loss for this network is incorporated as an additional term used when training the LM.  The authors show that by adding this loss term they can achieve SOTA (for single softmax model) perplexity on a number of LM benchmarks.

The technical contribution is proposing a new loss term to use when training a language model.  The idea is clear, simple, and well explained, and it seems to be effective in practice.  One drawback is that it is highly specific to language models.  Other recent works which have demonstrated effective regularization of LSTM LMs have proposed methods that can be used in any LSTM model, but that is not the case here.  In addition, there is not much theoretical justification for it, it seems like a one-off trick.  The loss term is motivated by the idea that we want the output distribution to retain some information about the context, but why should that be the case?

Although it is specific to language models, there are a few reasons it might be of broader significance:
- It falls in the recent line of work in incorporating auxiliary losses for various tasks.  This idea has touched many problems and seen success in practice.
- Perhaps it can be applied to other sequence models.  For example in encoder-decoder models, the decoder can be thought of as a conditional LM.

Experiments are comprehensive and rigorous.  They might be more convincing if there were results on a very large corpus such as 1 billion word corpus.

Pros:
- New SOTA for single softmax model on LM benchmarks.
- Simple, clearly explained idea.
- Demonstrates effectiveness of auxiliary losses.
- Rigorous experiments.

Cons
- Trick is specific to LM.
- No large corpus results.

---

> ### Author Response · Authors · 2018-11-20
> **Re: Weak accept**
>
> We thank the reviewer for a careful reading of the paper and the constructive comments. Although we proposed Past Decode Regularization (PDR) with language modeling in mind to exploit the symmetry between the input and output vocabulary (and the corresponding word embedding and softmax layer), any model/task that has this symmetry can potentially use a PDR term. As suggested by the reviewer, models for tasks like text summarization and neural machine translation (using a byte-pair encoding vocabulary as in Ofir & Wolf 2016) that use an encoder/decoder seq2seq architecture can benefit from PDR and is a topic of future research. We will incorporate this discussion in the updated version of the paper.
>
> We can justify PDR theoretically as an inductive bias on the language model. The observed bigrams in a language are not random and the distribution of the second word given the first word in a bigram is not uniform. Similarly, the distribution of the first word given the second word will be far from uniform. A RNN based language model models the first dependence (and more long range ones) and our proposed PDR tries to model the second one. In a unidirectional language model, we cannot look into the future tokens and hence we use the output distribution as a proxy for the "true second word" and decode the distribution of the first word. Thus the PDR term can be thought of as biasing the language model to retain more information about the distribution of the first word given the second word in a bigram.
>
> Finally, we have conducted further experiments on larger corpora, specifically the Gigaword corpus. We use a 2-layer LSTM with a word embedding dimension of 1024 and hidden dimension of 1024. We truncated the vocabulary by keeping approximately 100k words with the highest frequency. We compare the performance of the model with and without PDR and using no other regularization. We used the same validation and test sets as (Yang et al. 2017). We obtained a valid/test perplexity of 44.0/42.5 for the model with PDR and 44.3/43.1 for the model without PDR, showing a gain of 0.6 in the test perplexity by using PDR. We will incorporate these results in the experiments section and post the updated manuscript shortly.
>
>
> Press, Ofir, and Lior Wolf. "Using the output embedding to improve language models." arXiv preprint arXiv:1608.05859 (2016).
> Yang, Zhilin, et al. "Breaking the softmax bottleneck: A high-rank RNN language model." arXiv preprint arXiv:1711.03953 (2017).

---

### Public Comment · ~Adji_Bousso_Dieng1 · 2018-10-11
**Sub 50 perplexity on PTB has been achieved a while ago. Missing reference.**

Hi,

I wanted to point out one wrong claim from the paper. The current SOTA language model on the Penn Treebank (word level) is the model of Yang et al., 2017 [1] (faster convergence with Noisin regurlarization [2]).

One remark: have you looked into Pointwise Mutual Information? It seems as though you can interpret the regularizer you propose as enforcing a greater PMI between elements of the sequence. See for example [3] for use of PMI in conversation models to increase diversity in the generated responses.

[1] https://arxiv.org/abs/1711.03953
[2] https://arxiv.org/abs/1805.01500
[3] https://arxiv.org/abs/1510.03055

---

> ### Author Response · Authors · 2018-10-12
> **Regarding Sub 50 perplexity on PTB and PMI approach**
>
> Hi,
>
> Thank you for reading the paper and giving constructive comments.
>
> We are very much aware of the work of Yang et al., 2017 on breaking the softmax bottleneck and their results on language modeling. In fact we explicitly mention in section 4.1, line 6,7 that our result is the first sub-50 perplexity on PTB without the use of multiple softmaxes and add Yang et al. as a reference. We agree that this qualification should have been put in the abstract as well, and we will do so in the final version of the paper. As such, our Past Decode Regularization (PDR) method is orthogonal to the use of multiple softmaxes and can also be applied to such models as well.
>
> Thank you for pointing out the PMI paper, which we were not aware of. On preliminary study, apart from being applied to different problems (language modeling vs. sequence generation), we can observe several differences between their approach and ours. In their formulation, they use $p(S|T)$ as a measure to rerank responses (see section 4.2) at test time, where the candidate $T$ are themselves generated by ranking the outputs according to $p(T|S)$.  If we let $S$ to be the current token and $T$ to be the next token to pose language modeling in their framework, we do not compute $p(S|T)$ for any discrete token $T$, rather we use the predicted distribution on the next token $T$ to obtain a distribution over $S$. The backward probability we compute is not explicitly conditioned on $T$, rather it is conditioned on a distribution over the tokens that the LSTM predicts for $T$.  This also allows our method to be trained end to end, rather than being used as an adhoc reranking measure during test time.
>
> We believe our method is cleaner and a more effective way of biasing the LSTM to have more fidelity about past tokens. It would be interesting to see how an an algorithm that follows the approach presented in the paper performs on the language modeling task. The Twin Networks approach proposed in https://arxiv.org/pdf/1708.06742.pdf is also a closely related work. We will add this discussion to the final version of the paper.

---

> > ### Public Comment · ~Adji_Bousso_Dieng1 · 2018-10-16
> > **Making misleading statements in the abstract is bothersome.**
> >
> > It is even more bothersome now that you knew about Yang et al. [1] and still made a misleading statement in the abstract. You should not have waited until section 4.1 to state that your method is not state-of-the-art.
> >
> > [1] https://arxiv.org/abs/1711.03953

---

> > > ### Author Response · Authors · 2018-10-16
> > > **Disagree with your statement!**
> > >
> > > We respectfully but strongly disagree with your statement. We have no intention to mislead anybody. The work of Yang et al. is now well known in the language modeling world and a distinction is always made between models that use multiple softmaxes and those that don't, with the implicit understanding that the use of multiple softmaxes can lead to a further boost in performance.

---

> > > > ### Public Comment · ~Adji_Bousso_Dieng1 · 2018-10-16
> > > > **RE: Disagree with your statement!**
> > > >
> > > > Thank you for your reply. I am not sure I understand what you disagree with. You disagree that you should make it clear in your abstract that your method is not the "first sub-50 word perplexity on PTB"? That is the claim i have a problem with because it is misleading. The work does not need to claim SOTA to be interesting.

---

> > > > > ### Author Response · Authors · 2018-10-17
> > > > > **Got your point!**
> > > > >
> > > > > Thanks for the clarification! I do get your point and will definitely update the abstract in the final version to make things clearer.

---

> ### Public Comment · (anonymous) · 2018-10-21
> **Another related work**
>
> Hi,
>
> Here is also a NIPS 2018 paper related to the work. https://arxiv.org/pdf/1809.06858.pdf

---

> > ### Author Response · Authors · 2018-10-22
> > **Re: Another related work**
> >
> > Hi,
> >
> > Thanks for pointing this out. Will surely reference this in the updated version of the paper.

---

### Author Response · Authors · 2018-11-24
**Revised version uploaded**

We have uploaded a revised version of the paper. The main changes in brief, which have been discussed in detail in the comments below, are as follows

1. Addition of PDR to the Mixture-of-Softmaxes model of (Yang et al. 2017) produces further improvements for both PTB and WikiText-2 over the perplexities achieved by Yang et al. 2017.

                                                        AWD-LSTM-MoS+PDR  || AWD-LSTM-MoS (Yang et al. 2017)
Penn Treebank with finetuning -                   56.2/53.8   ||  56.5/54.4
Penn Treebank with dynamic evaluation -   48.0/47.3   ||  48.3/47.7

WikiText-2 with finetuning -                            63.0/60.5   ||  63.9/61.5
WikiText-2 with dynamic evaluation -            42.0/40.3   ||  42.4/40.7

The results have been added to Section 4.

2. PDR applied to a baseline LSTM model and trained on the Gigaword dataset gives modest improvements. We obtained a valid/test perplexity of 44.0/42.5 for the model with PDR and 44.3/43.1 for the model without PDR, showing a gain of 0.6 in the test perplexity by using PDR. The results have have been added to Section 4.

3. Section 3 has been condensed while more discussion has been added to Section 7 (related work). We also discuss the applicability of our PDR to other tasks which can be solved using seq2seq models and a shared vocabulary between the inputs and outputs.

4. A few typos have also been corrected. We have also updated the abstract to remove confusion about "state-of-the-art" and clearly specify the two cases of single softmax and mixture-of-softmaxes.

References
Yang, Zhilin, et al. "Breaking the softmax bottleneck: A high-rank RNN language model." arXiv preprint arXiv:1711.03953 (2017).

---

### Meta-Review · Area_Chair1 · 2018-12-14

**Confidence:** 3
**Recommendation:** Reject

**Metareview:**

The paper proposes an additional module to train language models, adding
a new loss that tries to predict the previous token given the next one, thus
enforcing the model to remember the past. Two out of 3 reviewers recommend to
accept the paper; the third one said it was misleading to claim SOTA since
authors didn't try the mixture-of-softmax model that is actually currently SOTA.
The authors acknowledged and modified the paper accordingly, and added a few
more experiments. The reviewer still thinks the improvements are
not important enough to claim significant novelty. Overall, I think the idea is simple and
adds some structure to language modeling, but I also concur with the reviewer about
limited improvements, which makes it a borderline paper. When
calibrating with other area chairs, I decided to recommend to reject the paper.